



# Technical note: Use of PM2.5 to CO ratio as a tracer of wildfire smoke in urban areas

Daniel A. Jaffe[1,2], Brendan Schnieder[3] and Daniel Inouye[3]

[1]School of STEM, University of Washington, Bothell, WA, 98011, USA
[2]Department of Atmospheric Sciences, University of Washington, Seattle, WA, 98195, USA
[3]Washoe County Health District, Air Quality Management Division, Reno, NV, USA

*Correspondence to*: Daniel A. Jaffe (djaffe@uw.edu)

**Abstract.** Wildfires, and the resulting smoke, are an increasing problem in many regions of the world. However, identifying
the contribution of smoke to pollutant loadings in urban regions can be challenging at lower concentrations due to the presence of the usual array of anthropogenic pollutants. Here we propose a method using the difference in PM to CO emission ratios between smoke and typical urban pollution. For smoke, emission ratios of PM2.5 to CO are between 200-300 $\mu g\ m^{-3}\ ppb^{-1}$, whereas typical urban sources have an emission ratio that is lower by a factor of 4-10. This gives rise to the possibility of using this ratio as an indicator of smoke extent. We use observations a regulatory surface monitoring sites in Sparks, NV, for
the period of May-September 2018-2021. During this time, there were many smoke-influenced periods from numerous California wildfires that burned during this period. Using a PM/CO ratio of 30, we can split the data into smoke-influenced and no-smoke periods. We then develop a Monte Carlo simulation, tuned to local conditions, to derive a set of PM2.5 /CO values that can be used to identify smoke influence in urban areas. From the simulation, we find that a smoke enhancement ratio of 140 $\mu g\ m^{-3}\ ppb^{-1}$ best fits the observations, which is significantly lower than the ratio observed in fresh smoke plumes.
The most likely explanation for this difference is greater loss of PM2.5 during dilution and transport to warmer surface layers. We find that the PM2.5/CO ratio in urban areas is an excellent indicator of smoke and should prove to be useful to identify biomass burning influence on the policy relevant concentrations of both PM2.5 and O3. Using the results of our Monte Carlo simulation, this ratio can also quantify the influence of smoke on urban PM2.5.

## 1. Introduction

In the U.S., smoke has become an increasingly challenging problem due to a significant increase in the area burned by wildfires (Zhuang et al 2021; Kalashnikov et al 2022; McClure and Jaffe 2018). Data from the National Interagency Fire Center (www.nifc.gov) show that between the early 1980s and 2021, the decadal average annual area burned by wildfires in the U.S. has increased by almost a factor of 3, from 1.1 to 3.0 million ha per year. Multiple factors are responsible for this increase, including climate change, increasing human ignitions and past forest management (Jaffe et al 2020).

Primary emissions from fires include fine particulate matter with a diameter of less than 2.5 $\mu$m (PM2.5), carbon monoxide (CO), nitrogen oxides (NOx=NO+NO2), and hundreds of volatile organic compounds (VOCs), including many toxic and



hazardous air pollutants (Akagi et al 2011; Permar et al 2021). In addition, secondary chemistry leads to $O_3$ and other secondary products. The cumulative impact of these emissions has substantial health implications (e.g., Ebi et al 2021; O'Dell et al 2020; 2021; Gan et al 2020; Doubleday 2020; Sorenson et al 2021). Smoke at the surface can be transported from nearby

or distant fires (e.g. DeBell et al 2004; Jaffe et al 2004; Teakles et al 2017; Rogers et al 2020). Satellites can provide an exceptional geospatial view of fires and the occurrence and transport of smoke (e.g. Duncan et al 2014; Jaffe et al 2020; Kahn 2020; O'Neil et al 2021; Holloway et al 2021). But with very few exceptions, satellite data provide little to no vertical information directly. Modeling of smoke transport and exposure is challenging for a number of reasons, including uncertainties in emissions, plume injection heights and model resolution ( Lu 2016; O'Neill 2021; Ye 2021). It is possible to

measure unique smoke markers, such as acetonitrile ($CH_3CN$) (Singh et al 2012; Chandra et al 2020), but these measurements are not routinely performed at surface sites, and even common tracer like acetonitrile also have some anthropogenic sources (Huangfu et al 2022).

Wildfire emissions are chemically distinct from anthropogenic, industrial or vehicular emissions in having very high $PM_{2.5}$ emissions per unit of fuel burned. For the U.S. as a whole, the U.S. EPA reports a $PM_{2.5}$ to CO emission ratio (ER) of 0.085

in 2017 (g/g) for all emission sources, excluding wildfires (U.S. EPA 2022). This ratio drops to 0.076 if residential wood combustion is also excluded. Akagi et al (2011) report an emission ratio of 0.12-0.14 (g/g) for temperate and extra-tropical wildfires. In urban areas, it is likely that this ratio is even lower, due to a greater contribution from mobile sources (cars and trucks), which have a $PM_{2.5}$ to CO emission ratio of 0.009 (g/g). Observations of $PM_{2.5}$ and CO in urban areas often show good correlation and these can be used to derive the normalized enhancement ratio (NER, $\Delta PM_{2.5}/\Delta CO$), which reflects the

emissions, chemical and physical processing, and any background contribution. Reported $\Delta PM_{2.5}/\Delta CO$ NERs in urban areas from several studies are in the range of 21-45 $\mu g\ m^{-3}\ ppm^{-1}$, which would correspond to an emission ratio of 0.018-0.039 g/g, assuming no loss of either species (Dimitriou and Kassomenos, 2014; Patton et al., 2014). Laing et al (2017) looked at the $PM_{2.5}$ to CO correlation in 8 western U.S. cities for non-smoke periods and found reasonable correlations ($R^2$ values of 0.32-0.57) and slopes of 21-66 with a median value of 35 $\mu g\ m^{-3}\ ppm^{-1}$ (corresponding to emission ratios of 0.018-0.057 g/g).

Laing et al (2017) also found that smoke-influenced periods had much higher $PM_{2.5}$ to CO correlation slopes ranging from 57-228, with a median of 128 $\mu g\ m^{-3}\ ppm^{-1}$ (n=25). Garofolo (2019) reported values of 80-400 $\mu g\ m^{-3}\ ppm^{-1}$, with most between 200-300 for the organic aerosol to CO NER for 20 fires during the 2018 WECAN experiment. Briggs et al (2016) report mean aerosol scattering to CO NERs of 0.90 for 23 plumes at MBO, corresponding to a PM/CO ratio of 200-300 $\mu g\ m^{-3}\ ppm^{-1}$, depending on the mass scattering efficiency and Kleinmann et al (2020) report mean values of 317 and 361 $\mu g\ m^{-3}\ ppm^{-1}$ for

aged and fresh plumes, respectively, during the 2013 BBOP experiment. Selimovic et al (2019; 2020) note that the PM/CO NER in ground-level smoke is about half of that observed from aircraft or free tropospheric observations. This is most likely caused by evaporation of organic aerosol mass due to higher surface temperatures and greater downstream dilution. These past observations present a fairly consistent picture showing that $PM_{2.5}$/CO NER for surface smoke is about 3-4 times greater than the NER for typical urban observations in the absence of smoke, based on the medians in Laing et al 2017.





The very different PM$_{2.5}$ to CO NERs for typical urban air and smoke events suggest that the observed ratios can be used to derive the smoke contribution to surface PM$_{2.5}$ concentrations. To examine this hypothesis, we use data from a monitoring site in Sparks, NV, near Reno, a region that has been heavily influenced by smoke in the past several years due to the large number and extent of California wildfires. Data from this region were recently used to examine the role of high PM$_{2.5}$ exposure from smoke on COVID-19 incidence (Kiser et al 2021). From the Sparks, NV observations, we develop a quantitative model

using a Monte Carlo simulation that provides a range of probabilistic results that can be compared to observations. We find that this method appears to reasonably quantify the smoke contribution in an urban area.

## 2.   Methods and data sources

For this analysis, we use daily mean PM$_{2.5}$ and CO concentrations for May-September 2018-2021 from the Sparks, NV routine air quality monitoring site (EPA AQS identification #320311005) near Reno, NV that is operated by the Washoe (NV) County

Health District, Air Quality Management Division. The site uses instruments and standards that are consistent with the national EPA requirements (40 CFR Part 58) and report data into the EPA's national Air Quality System (AQS). The Sparks site has near-continuous measurements of PM$_{2.5}$, CO and O$_3$. We use data for May–September 2018–2021 to avoid complications with sources from residential wood combustion. Data were obtained from the EPA AirData site (https://www.epa.gov/outdoor-air-quality-data), except for 2021 data, which were obtained from AirNow-Tech, a web-based data resource operated for the

U.S. EPA (https://www.airnowtech.org/). We note that 2021 data is considered preliminary at this time, although in practice the preliminary data usually do not change significantly. Instrumentation at the Sparks site include a MetOne model 1020 Beta Attenuation Monitor (BAM) for PM$_{2.5}$, a Teledyne API model 300 EU non-dispersive IR monitor for CO and a Teledyne API model T400 UV O$_3$ analyser. These instruments have stated detection limits (DLs) of 1 µg m$^{-3}$, 20 ppb and 0.4 ppb, respectively. Because there were some zero and very low values for PM$_{2.5}$ any concentration less than the DL was set to 1 µg

m$^{-3}$. This impacted less than 2% of the dataset. No below DL values were reported for the CO or O$_3$ data. As an indication of overhead smoke, we use the daily product from the NOAA Hazard Mapping System-Fire and Smoke Product (hereafter simply HMS). This product is based on analysis of multiple satellite products, both geostationary and polar orbiting. More details on HMS are in Rolph et al (2009) and Kaulfus et al (2017). We note that HMS can sometimes miss thin smoke plumes, especially in the presence of clouds (Buysse et al 2019). Buysse et al (2019) found that there is enhanced surface

PM$_{2.5}$ on 30-70% of the days with overhead HMS smoke, depending on the location.

## 3. Results

Figure 1 shows one example of the HMS smoke product for the Loyalton fire on Aug. 16, 2020, which was about 35-45 km from the Sparks monitoring site. This fire started on 8/14/2020 and burned for approximately one month. In total, this fire

burned approximately 20,000 ha in the Tahoe and Humboldt-Toiyabe National Forests. On 8/16/2020, the daily mean PM$_{2.5}$





and CO concentrations were 38 µg m$^{-3}$ and 0.43 ppm at the Sparks, NV monitoring site.  As Washoe County is located just east of the California-Nevada border, smoke from many fires in California is often transported to the Sparks monitor.  Table 1 shows data for the number of days that exceeded the U.S. National Ambient Air Quality Standards (NAAQS) for PM$_{2.5}$ (2006, 24-hour standard, daily mean of 35 µg m$^{-3}$) and O$_3$ (2015 8-hour O$_3$ standard, maximum daily 8-hour mean of 0.070 ppm) for

the Sparks monitoring site, along with the annual area burned in California.  While 2020 was the highest year on record for the area burned in CA for the past 2 decades, 2021 was the second highest year and had a greater number of days in Reno that exceeded the NAAQS.   Note that 2019 was a particularly low fire year in CA and there were no exceedances of either the daily PM$_{2.5}$ or O$_3$ NAAQS at the Sparks monitoring site.   Overall, for this time period (May-September 2018-2021), 200 out of 612 days had overhead HMS smoke  at the Sparks monitoring location.   The PM$_{2.5}$/CO smoke criteria is discussed later in

this section.

| | 2018 | 2019 | 2020 | 2021 |
|---|---|---|---|---|
| California area burned (Ha) | 7.4E5 | 1.0E5 | 1.7E6 | 1.1E6 |
| Sparks Overhead HMS smoke (days) | 51 | 11 | 52 | 86 |
| Sparks smoke days* | 30 | 5 | 64 | 57 |
| PM$_{2.5}$ exceedance days | 6 | 0 | 19 | 22 |
| PM$_{2.5}$ exceedance days with smoke* | 6 | 0 | 19 | 22 |
| O$_3$ exceedance days | 10 | 0 | 5 | 13 |
| O$_3$ exceedance days with smoke* | 10 | 0 | 5 | 11 |

Table 1: California area burned, overhead HMS smoke days, and days over the U.S. National Ambient Air Quality Standard at Sparks, NV for PM$_{2.5}$ (daily mean of 35 µg m$^{-3}$ ) and O$_3$ (70 ppb, 8 hour average).   The smoke criteria (indicated by *)  uses a PM$_{2.5}$/CO ratios of 35, as discussed later in text.

Figure 2 shows the daily PM$_{2.5}$ vs CO concentrations for May-Sept 2018-2021, segregated for smoke vs non-smoke conditions.  The data are segregated using (i) the HMS smoke product and (2) a PM$_{2.5}$/CO ratio greater or less than 30.  The value of 30 is chosen based, in part, on the work of Laing et al (2017) and on evaluation of likely smoke influence.   We find the slopes and correlations are not strongly influenced by the choice of PM$_{2.5}$/CO ratio.   For example, using a ratio of <20, 30, 40 and 50 we get slopes of 16.5, 18.1, 23.4 and 33.9 µg m$^{-3}$ per ppm, an increasing pattern as would be expected.   We found that smoke

influence can be observed on some days at a PM/CO ratio as low as 32.  An example of this is 8/5/2018, when extensive and heavy smoke blankets most of California, Nevada and other western states.  PM$_{2.5}$ and CO concentrations at Sparks are 22 µg m$^{-3}$ and 0.68 ppm, respectively, for a PM$_{2.5}$/CO ratio of 32.    So, while the relatively low ratio implies significant mixing of this smoke event with air containing a lower ratio,  the high PM$_{2.5}$ concentrations and widespread smoke is consistent with a significant smoke influence on that day.   Using the PM$_{2.5}$/CO ratio to segregate the data, we find an improved correlation of

PM and CO in the lower range of ratios, compared with using the HMS alone as an indicator (Figure 2).   Table 2 summarizes the dataset, as segregated by the PM/CO ratio as well as using the HMS smoke product alone.    While there is relatively little change in the mean and SD of the smoke-influenced and non-smoke data, the improved correlation suggests that the PM$_{2.5}$/CO



ratio is a better way to segregate the dataset. We note that the exact choice of PM₂.₅/CO ratio depends on the certainty required. This is discussed in more detail using a Monte Carlo simulation, as described below. We note that there are 53 days with
overhead HMS smoke, but a PM₂.₅/CO ratio<30 and 60 days with a PM₂.₅/CO ratio>30 and no HMS smoke.

**Table 2. Sparks daily PM₂.₅ and CO data for May-September 2018-2021, segregated by the PM₂.₅/CO ratio and by overhead HMS smoke.**

|  | PM₂.₅/CO <30.0 (no smoke) | PM₂.₅/CO >30.0 (smoke-influenced) |
|---|---|---|
| **Count** | 414 | 198 |
| **Mean PM₂.₅ (µg m⁻³)** | 4.8 | 27.7 |
| **Std. Dev. (µg m⁻³)** | 1.8 | 29.2 |
|  | **HMS=0 (no smoke)** | **HMS=1 (smoke-influenced)** |
| **Count** | 412 | 200 |
| **Mean PM₂.₅ (µg m⁻³)** | 5.1 | 26.9 |
| **Std. Dev. (µg m⁻³)** | 1.9 | 29.6 |

We use the PM₂.₅ and CO data to develop a Monte Carlo simulation of the PM/CO ratio for Reno using the following
relationships:

$$\text{PM}_{2.5} \ (\mu g \ m^{-3}) = \text{Urban PM}_{2.5} + \text{Smoke PM}_{2.5} + \text{background PM}_{2.5} = 10^{\alpha} + 10^{\beta} + 2 \ \mu g/m^3 \qquad (1)$$

$$\text{CO (ppm)} = \text{Urban PM}_{2.5}/\text{R}_{urban} + \text{Smoke PM}_{2.5}/\text{R}_{smoke} + 0.2 \ \text{ppm} \qquad (2)$$

Where $R_{urban}$ and $R_{smoke}$ are the NERs ($\Delta PM_{2.5}/\Delta CO$) to represent urban emissions and smoke, respectively. The smoke terms in equations 1 and 2 are only included on 1/3 of the days, corresponding to the fractional incidence of HMS smoke.
We explore a range of values for $R_{urban}$ and $R_{smoke}$ as shown in Table 3. The parameters α and β are used to model the log-normal distributions for urban PM₂.₅ with, and without, smoke PM₂.₅, respectively. Equations 1 and 2 include a background contribution to represent natural, biogenic, and intercontinental sources of PM₂.₅ and CO. The background concentrations were set to 2 µg m⁻³ for PM₂.₅ and 0.2 ppm for CO. These background values were estimated based on observations from 2019, a low fire year, from a rural continental site (West Yellowstone, MT AQS #300310017) and a marine background site
(Cheeka Peak, WA, AQS #530090013). For the May-Sept 2019 period the West Yellowstone mean values for PM₂.₅ and CO were 2.5 µg m⁻³ and 0.24 ppm, whereas for the Cheeka Peak site the mean values were 2.1 µg m⁻³ and .08 ppm. Median values were very similar at both sites. While PM₂.₅ concentrations were similar at both sites, whereas CO was higher at the continental site. Given that Sparks, NV is a continental/inland location, the West Yellowstone, MT concentrations are likely more representative of its background concentrations.

The Monte Carlo simulations estimate the observed PM₂.₅ and CO concentrations, and the ratio, using Equations 1 and 2. The simulation computes 10,000 concentrations, where α, β, $R_{urban}$ and $R_{smoke}$ are allowed to vary independently with values as defined in Table 3. These values are chosen to be consistent with the mean and S.D. of the non-smoke (α) and smoke (β)





datasets, respectively, excluding the contribution from background concentrations. Note that the Monte Carlo simulations are intended to reflect the bulk distributions, so there is no correspondence between an individual day in the simulation with

any particular day in the observations.

**Table 3. Parameter values used in the Monte Carlo simulations. For the $R_{urban}$ and $R_{smoke}$ parameters, multiple mean values are considered.**

|  | α<br>(unitless) | β<br>(unitless) | $R_{urban}$<br>(µg m$^{-3}$ ppm$^{-1}$) | $R_{smoke}$<br>(µg m$^{-3}$ ppm$^{-1}$) |
|---|---|---|---|---|
| **Mean** | 0.4 | 1.3 | 20,40,80 | 100,140,200 |
| **Std. Dev.** | 0.2 | 0.4 | 10 | 20    160 |

Figure 3 shows results of the simulation and with varying mean values for the $R_{smoke}$ parameter. Even at very high PM$_{2.5}$ concentrations, the observed PM$_{2.5}$/CO ratio never exceeded 125 µg m$^{-3}$ ppm$^{-1}$. The simulation suggests an optimum $R_{smoke}$ value of 140 µg m$^{-3}$ ppm$^{-1}$. So, consistent with the work of Laing et al (2017) and Selimovic et al (2019; 2020), we find that

the best-fit NER values at the surface are much lower than emission factors reported for fresh or free tropospheric smoke plumes.

Figure 4 shows the results of the simulations with varying values for the $R_{urban}$ parameter. Here, the best value is more difficult to discern. At high PM$_{2.5}$ concentrations and PM/CO ratios, this parameter has very little influence on the simulated values. At the low range of PM$_{2.5}$ concentrations a value of 20 is clearly too low, but there is little difference between the other values

and so it is not clear which value is optimal. This parameter should reflect the primary PM$_{2.5}$ and CO emissions in the area, plus contributions from secondary organic aerosol (e.g. Nault et al 2021). For Washoe County, NV (the county containing Reno and Sparks) the EPA's 2017 National Emission Inventory gives primary emissions of PM$_{2.5}$ and CO of 1,482 and 55,529 short tons per year, excluding wildfires and residential wood combustion. This corresponds to a PM$_{2.5}$/CO emission ratio of 0.034 g/g or an enhancement ratio of 39 µg m$^{-3}$ ppm$^{-1}$. An important constraint on using this approach to discern the urban,

non-smoke PM$_{2.5}$/CO NER are limitations on the instrumentation and the impact of background concentrations at low PM$_{2.5}$ and CO concentrations. Nonetheless, we find that using an $R_{urban}$ parameter of either 40 or 80 has little influence on our results at higher PM$_{\mathbf{2.5}}$ concentrations. For the remaining of this analysis, we use an $R_{smoke}$ value of 140 and an $R_{urban}$ value of 40.

Figure 5 shows the fractional smoke contribution to PM$_{2.5}$ vs the PM$_{2.5}$/CO NER from the Monte Carlo simulations. As specified in the model setup, 2/3 of the points have no smoke contribution. These have a mean PM$_{\mathbf{2.5}}$/CO value of 17, with a

range of 6-34. As the Monte Carlo simulations represent a probabilistic approach, we can also look at the likelihood that a given set of points has a specific degree of smoke influence. Figure 5 shows the probability that a given set of PM$_{\mathbf{2.5}}$/CO ratios (binned in units of 10) has more than 50% of the PM$_{\mathbf{2.5}}$ due to smoke. So, starting with the PM$_{\mathbf{2.5}}$/CO bin of 30-40, we have a very high probability (0.82) of more than 50% PM$_{\mathbf{2.5}}$ due to smoke and at a bin of 40-50, we have near certainty (0.996).



We can use the information in Figure 5 to evaluate the likelihood that smoke contributed to the days with high PM$_{2.5}$ or O$_3$, as shown in table 1. The years 2018, 2020 and 2021 all had significant number of exceedances days (over the NAAQS), whereas the low fire year of 2019 had none. For Table 1, we use a PM$_{2.5}$/CO values of 35 which, based on the Monte Carlo simulation, implies that smoke contributes more than half of the total PM$_{2.5}$ on 85% of days. Even using a smoke criteria of PM$_{2.5}$/CO of 45, we find no change in the number of smoke influenced days. Not surprisingly, the PM$_{2.5}$/CO criteria identified all of the PM$_{2.5}$ exceedance days as smoke influenced, using either smoke criteria (35 or 45). For O$_3$, the results show that 24 out of the 26 exceedance days were smoke influenced, using either criteria. While the PM$_{2.5}$/CO ratio can quantitatively estimate the fraction of PM due to smoke (e.g. Figure 5), we note that this approach can not provide a quantitative estimate of the smoke contribution to the O$_3$ levels. Other tools would be needed to quantify the smoke contribution to the MDA8 O$_3$ values (e.g. Ninneman and Jaffe 2021; Jaffe 2021; Gong et 2017). Nonetheless, the results shown in Table 1 clearly show that the PM$_{2.5}$/CO can identify days with a strong smoke signature.

## 4. Summary

The large difference in emission ratios between typical urban pollution and wildfire smoke gives rise to very different observed NERs in urban areas for non-smoke and smoke-influenced conditions. We find that a Monte Carlo simulation of mixing between smoke and non-smoke NERs can accurately reproduce the observed NERs and provides a measure of smoke influence in an urban area. The model supports earlier work that finds the PM$_{2.5}$/CO NER in biomass burning influenced plumes at surface sites is approximately half of that observed in fresh emissions and in cooler environments. This likely is caused by loss of PM mass during transport due to dilution and warmer temperatures at surface sites. For the Sparks, NV monitoring site we find that at a PM$_{2.5}$/CO ratio of 35 $\mu$g m$^{-3}$ ppm$^{-1}$ biomass burning contributes more than half of the total PM$_{2.5}$ on 85% of days. These calculations are based on the observed concentrations at one regulatory monitor, but should be applicable to other sites. To apply the Monte Carlo simulation at other sites requires that the parameters in Table 3 be adjusted to fit the local data. The R$_{urban}$ parameter would need to be adjusted based on local emissions and observations and the $\alpha$ and $\beta$ parameters would need to be fit based on the observed non-smoke and smoke concentrations, respectively.

This analysis demonstrates that it is possible to identify smoke at the surface based on commonly measured air pollutants with high confidence. While satellite data can also identify smoke influence, these have a high false positive rate, meaning that many days identified by satellite products as having overhead smoke show little or no influence at the surface. We propose that the observed PM$_{2.5}$/CO ratio provides a more robust signal of surface smoke in urban areas and with no false positives.

### Acknowledgements

This work was supported by the National Oceanic and Atmospheric Administration (NOAA; grant number NA17OAR431001). The authors acknowledge helpful comments from Nathan May and Matthew Ninneman.





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



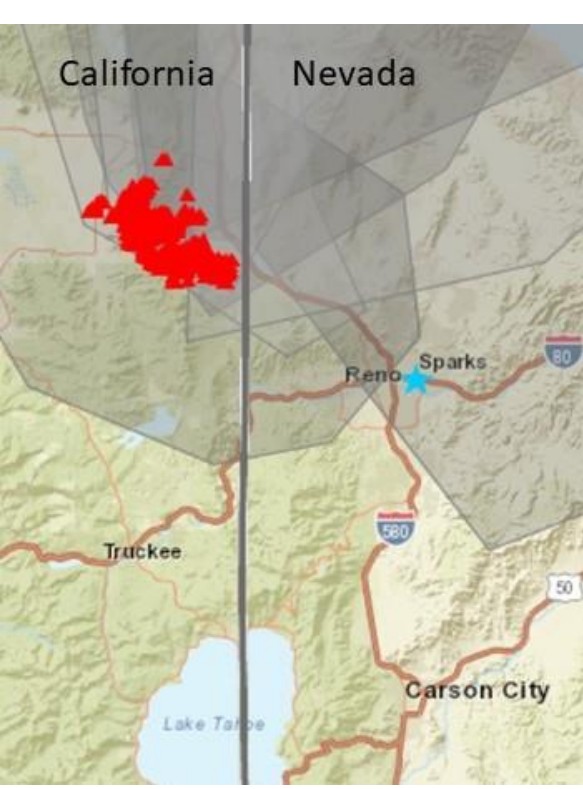


**Figure 1: NOAA HMS smoke and fire location for Aug. 16, 2020. The Loyalton fire is burning in California near the Nevada border at this time. The blue star shows the location of the Sparks, NV monitoring site, which is approximately 35-45 km from the fire. This map was created from the AirNowTech site (https://www.airnowtech.org/).**






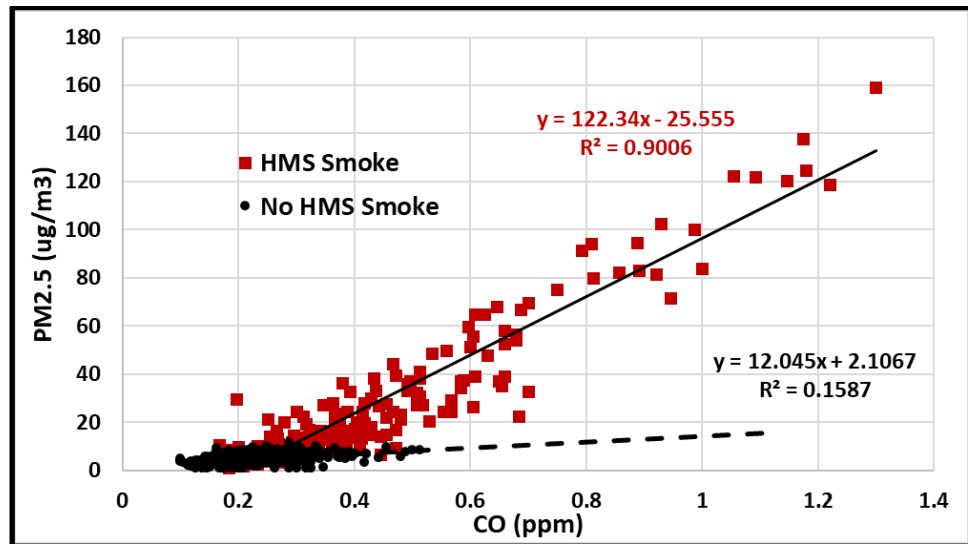

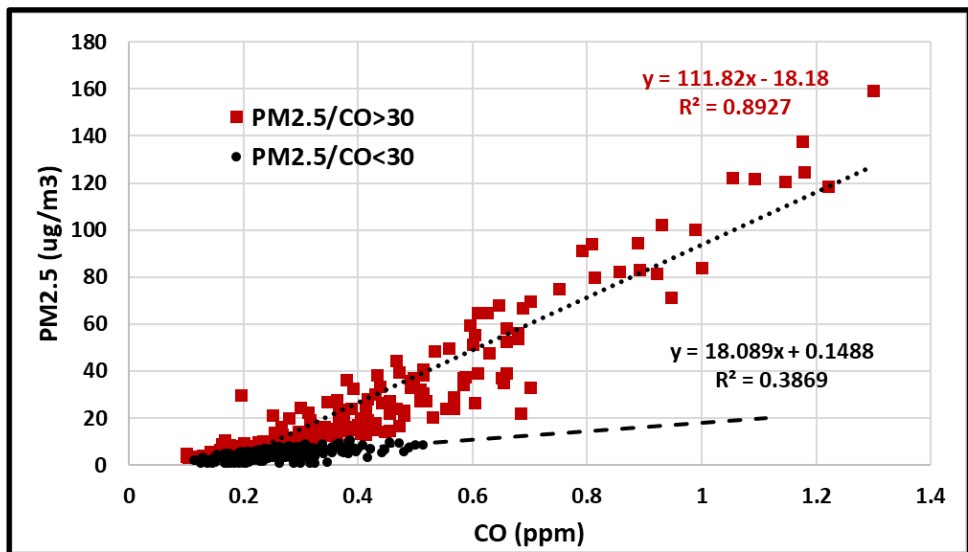

**Figure 2: Observed PM$_{2.5}$ vs CO for May-September data (May 1, 2018-August 31, 2021). Each point is the daily mean of observed values sorted by (a) overhead HMS smoke product or (b) PM/CO ratio of 30 μg m$^{-3}$/ppm.**





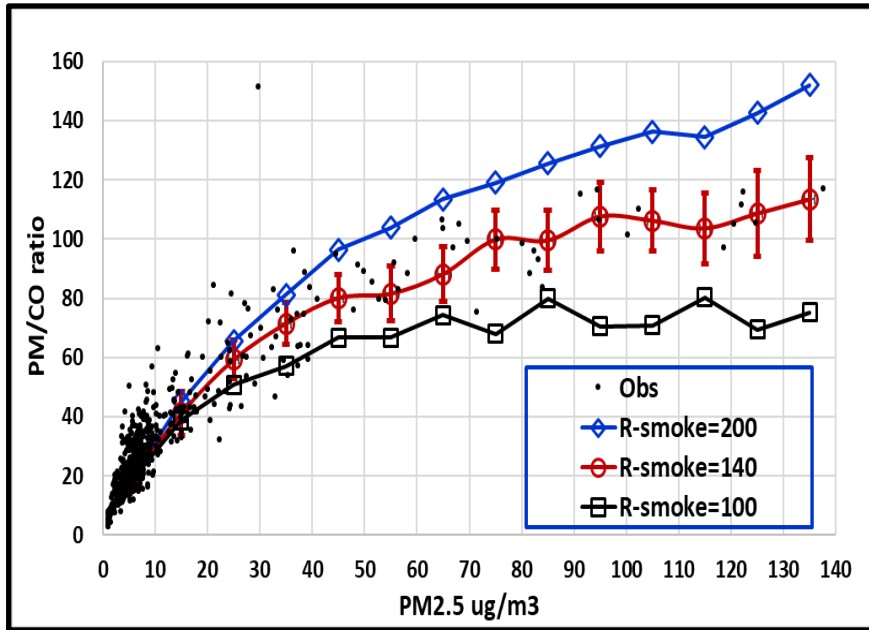

**Figure 3.** PM$_{2.5}$/CO ratio (µg m$^{-3}$ ppm$^{-1}$) vs PM$_{2.5}$. The black dots show the observations, and the black square, red circle and blue diamonds show the influence of the R-smoke parameter for the urban + smoke simulations. The simulation results are binned in 10 µg m$^{-3}$ intervals centered on the indicated values. For these simulations, R-urban is fixed at 40. Error bars show 1 σ on the middle simulation. One observation is not shown (PM/CO ratio of 122 and a PM$_{2.5}$ concentration of 159 µg m$^{-3}$).

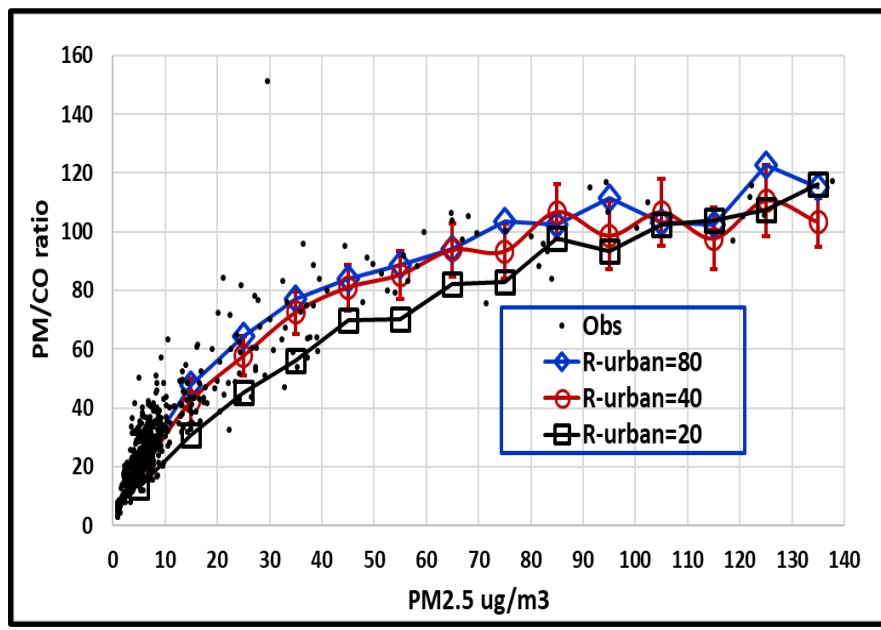

**Figure 4.** PM$_{2.5}$/CO ratio (µg m$^{-3}$ ppm$^{-1}$) vs PM$_{2.5}$. The black dots show the observations, and the black square, red circle and blue diamonds show the influence of the R-urban parameter on the Monte Carlo simulations. The simulation results are binned in 10 µg m$^{-3}$ intervals centered on the indicated values. For these simulations, R-smoke is fixed at 140. Error bars show 1 σ on the middle simulation. One observation is not shown (PM/CO ratio of 122 and a PM$_{2.5}$ concentration of 159 µg m$^{-3}$).

400

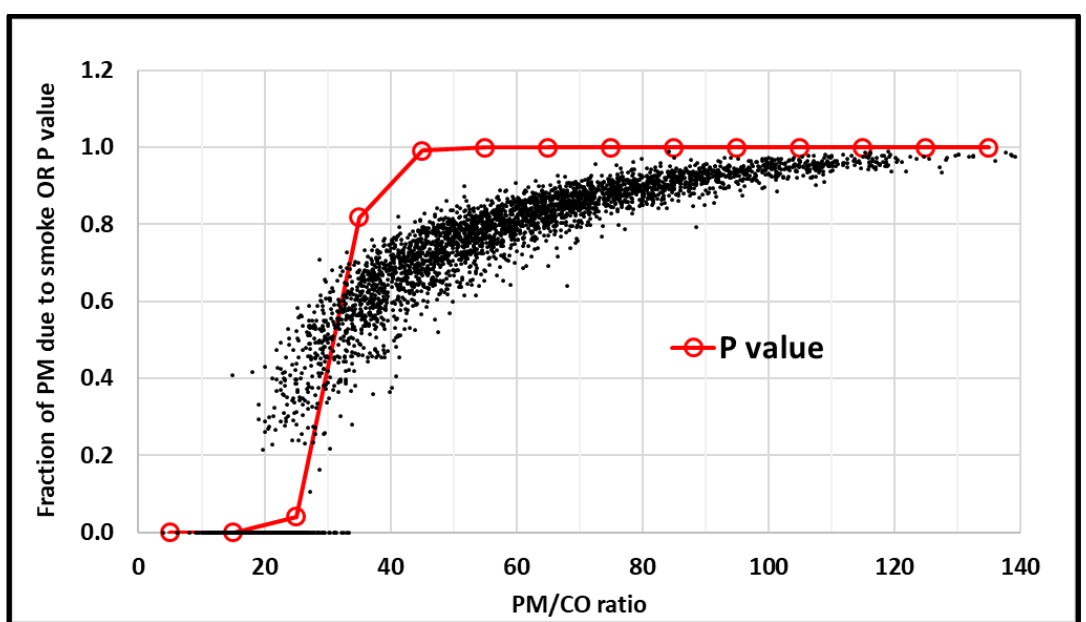

**Figure 5.** **Fraction of PM$_{2.5}$ due to smoke vs the PM$_{2.5}$/CO ratio (µg m$^{-3}$ ppm$^{-1}$) as calculated from the Monte Carlo**
405 **simulations. We note that the Monte Carlo simulations give a probabilistic relationship. So, for example, at a PM/CO**
**ratio of between 30 and 40, 83% of the points have more than half of the PM$_{2.5}$ due to smoke. The red open circles**
**show the probability that more than 50% of the PM$_{2.5}$ is due to smoke, within each PM/CO bin.**

410