# Peer review of "Technical note: Use of PM2.5 to CO ratio as an indicator of wildfire smoke in urban areas"

_Atmospheric Chemistry and Physics, 2022_

## Author Comment (AC1)

Thank you to the two reviewers for the positive feedback, detailed comments and occasional corrections. Below is our response to each comment (original in black, response in red).

RC1:

The manuscript is an excellent contribution to the scientific community for this subject. However, several technical corrections should be made for clarity and consistency. These are: 1) past work should be discussed in the past tense, not present tense; 2) PM2.5 should not be referred to as PM, since they are not equivalent (this problem appears in both text and figures); 3) "tracer" and "indicator" should not be used interchangeably. The authors should make sure the three listed items (above) are corrected for the entire manuscript. I have also suggested several technical corrections (below) that could help clarify the writing. Agree. These changes have been made in the manuscript.

Title: change "a tracer" to "an indicator"—I agree. This is a better title.

Abstract:

Line 9: remove commas Done

Line 10: change "lower" to "low Done

Line 11: PM2.5 Done

line 12: specify this is in regards to wildfire smoke Done.

Line 14: "extent" is ambiguous; temporal or spatial extent? Reworded.

Line 14: change to "We use observations at a regulatory surface monitor site..." Done

Line 15: remove "During this time," Done

Line 16: PM to PM2.5; change to "ratio threshold of 30" Done

Line 19: "in fresh smoke plumes" - provide example range of values Values added.

Line 23: change to "...this ratio can also help quantify..." Done

1.Introduction:

Line 25: change to "... smoke impacts have become more prevalent due to..." Original preferred

Line 30: change to "diameter less than" Done

Line 31: change to "...(VOCs) which include many toxic..." Done

Line 32: change "In addition, secondary" to "Furthermore, atmospheric" Done.

Line 33: change "emissions" to "pollutants" Original preferred

Line 34: New paragraph Done

Line 34: change "be transported" to "originate" Done

Line 40: change "markers" to "tracers" Done

Line 41: change to "... at surface sites and also have some anthropogenic..." Done

Line 43: change "anthropogenic" to something like "commercial, institutional, residential".  Otherwise it is a redundant statement since industrial and vehicular are already anthropogenic Done

Line 43: change "in having" to "with" Original preferred

Line 44: remove "For the US as a whole," Original preferred

Line 45: change to "... emission sources in the U.S., excluding..." Original preferred

Line 46: change "report" to "reported" Done

Lines 43 to 64:  this paragraph could be summarized in a table. **This is a good suggestion.   The paragraph will get shortened and the information summarized in a new table.  See below:**

**Table 1.   Emission ratios and observed NERs for non-smoke and smoke conditions.**  **Emission ratios are converted into NERs using a pressure of 1 atmosphere and temperature of 272K (STP).  This calculation assumes no loss of either $PM_{2.5}$ or CO.  For observed NERs, the study mean is given and range (if known) is shown in parentheses.**

| | $PM_{2.5}$/CO E.R. (g/g) | $PM_{2.5}$/CO NER ($\mu$g m$^{-3}$ ppm$^{-1}$) |
|---|---|---|
| **Non-smoke emissions and observed NERs** | | |
| US industrial and mobile emissions (excludes wildfires and residential wood combustion)[1] | 0.076 | 95 |
| U.S. Mobile sources only[1] | 0.009 | 11 |
| Observed NERs in urban areas with no smoke[2] | | 37 (21-66) |
| **Smoke emissions and observed NERs** | | |
| Temperate wildfires ERs[3] | .142 | 177 |
| Temperate wildfires ERs[4] | .176 (.07-.57) | 220 (87-712) |
| Observed smoke NERs in urban areas[2] | | 128 (57-228) |
| Observed smoke NERs, surface sites[5] | | 103 (120-156) |
| Fresh plumes, high elevation site[6] | | 258 (66-377) |

| | | |
|---|---|---|
| Fresh plumes, high elevation site and aircraft data[7] | | 299 (170-630) |
| Fresh plumes, aircraft data[8] | | 201 (80-400) |
| Fresh plumes, aircraft data[9] | | 339 (21—492) |

[1]Data from the EPA's 2017 National Emission Inventory (EPA 2022).
[2]Data from Laing et al 2017.
[3]Data from Akagi et al 2011.
[4]Data from Anderea 2019.
[5]Data from Selimovic et al 2020.
[6]Data from Briggs et al 2016.   Scattering values are reported at STP and converted to $PM_{2.5}$ using a dry mass scattering coefficient of 3.5 $m^2g^{-2}$
[7]Data from Collier et al 2016.   This value includes refractory $PM_1$. Values are reported at STP.
[8]Data from Garofolo et al 2019.  This value includes only the organic, non-refractory fraction, however this is likely more than 90% of total $PM_{2.5}$ mass.
[9]Data from Kleinman et al 2020.  This value includes only the non-refractory $PM_1$ mass.

\*\*\*\*\*\*\*\*\*\*\*\*\*\*\*\*\*\*\*\*

Line 65: change to "...smoke events should allow us to use the observed ratios to derive the smoke..." Original preferred

Line 68: remove "recently" Done

Line 69: change "develop" to "developed" Done

Line 70: include a reference to what a Monte Carlo simulation is Done.

*Baez, J.C. and Tweed, D. Monte Carlo methods in Climate Science.  Math Horizons, November 2013.  (available at: https://www.maa.org/sites/default/files/pdf/horizons/baeztweed_nov13.pdf, accessed May 2022.)*

2. Methods and data sources:

Line 73: remove "routine" Done

Line 74: remove (NV) Original preferred

Line 80: preliminary data should be finalized and this sentence should be removed Done.

Line 84: remove "Because there were some zerio and very low values for" Original preferred

Line 84: change to "PM2.5 concentrations less than the DL were set..." Done

Line 86: change to "...we use the daily smoke polygons from..." Done

Line 87: change to "The smoke polygon product is created by expert image analysts that digitize smoke plume extent a few times per day based on analysis of GOES-16 and GOES-17 ABI True Color Imagery available during daylight hours." Done.

3. Results:

Line 96-97: change to "Washoe County is located due east of the California-Nevada border, so smoke from fires in California..." Done

Line 98: remove comma at end of line Done

Table 1 caption last sentence: "ratios" should not be plural; "35" should include units Done

Line 111: change "(i)" to "(1)" Done

Line 115: change PM to PM2.5 Done

Lines 116 and 117: both sentences should be changed to past tense Done

Line 117: remove "So, while" Done

Line 118: change to "... lower ratio, but the large PM2.5 concentrations..." Done

Line 120: correct PM to PM2.5 and remove "alone" Done

Line 121: correct PM to PM2.5; change "as well as using" to "and"; change "alone" to "separately". Done

Line 122: change "change" to "difference" Done; change "smoke influenced and non-smoke data" to "segregated data methods" Original preferred

Line 123: remove "We note that" Done

Line 124: change "are" to "were" Done

Line 134: change "use" to "used" Done

Line 139: change "are only included" to "were non-zero" Done

Line 140: change "model" to "represent" Done

Line 145: change "For" to "During" Done

Line 147: remove "whereas" Original preferred

Line 162: remove "and" Done

Line 165: include values for comparison Done.

Line 167: remove "Here,"; change "is more" to "of R-urban was" Done

Line 168: change PM/CO to $PM_{2.5}/CO$; change "this parameter" to "R-urban" Done

Line 169: include units for value of 20 Done

Line 170: remove first occurrence of "and" Done

Line 172: change Emission to Emissions Done

Lines 174-176: This sentence need to be consistent with plural and singular usage; change "this approach" to "the Monte Carlo approach" Done

Line 177: change "remaining" to "rest" Done; move "for the rest of this analysis" to end of sentence (and remove comma). Original preferred

Line 185: change "all had significant" to "all had a significant" Done

Line 191: change PM to PM2.5 Done

4.Summary:

The summary should specify that the conclusions are specific to warm weather (or May-Sept) where RWC is not a factor. Other details about the monitoring details could also be shared. Done.

The final paragraph should have at least one instance of "smoke" changed to "wildfire smoke". Last sentence: change "propose" to "conclude". Done

Figure 2: change "sorted" to "segregated"; two graphs in one figure should be denoted as (a) and (b) on the graphs or referred to as top/bottom in the text; PM to PM2.5 in text Done

Figure 3: R-smoke should use subscript; "Monte Carlo" should come before "simulations". Done.

Figure 4: R-smoke should use subscript Done.

Figure 5: PM to PM2.5 (text and figure) Done.

RC2:

This manuscript describes a new method to determine the relative contribution of smoke to observed $PM_{2.5}$ during wildland fire smoke season. Following Liang et al, 2017, the authors use $PM_{2.5}/CO$ to categorize smoke and non-smoke influenced days. In contrast to the overhead HMS

smoke product from satellite measurements that can misrepresent conditions on the ground due to inadequate (or nonexistent) retrieval of near-surface smoke concentrations, the $PM_{2.5}/CO$ method uses in-situ ground measurements typically present at regulatory surface monitoring sites. After determining the $PM_{2.5}/CO$ ratios for urban and smoke aerosol by comparing Monte Carlo simulations to observations, the authors estimate relative contribution of smoke to $PM_{2.5}$ for smoke-influenced days, finding that indeed all the $PM_{2.5}$ exceedance days during the period of study have high influence of smoke. Because the simulation is trained on local conditions, the values reported here may not be widely applicable, but the method can be applied to other sites to identify and estimate relative smoke influence. This manuscript describes the development of methods for interpretation of atmospheric data, but with a limited scope of one study location where all PM2.5 exceedance days are from smoke, so its publication as a Technical Note is appropriate. I recommend publication with minor revisions below:

1. Discussion of previous work may be improved by description of the various units for normalized enhancement ratios (NERs).
   2. $\Delta PM_{2.5}/\Delta CO$ in g/g vs ug m$^{-3}$ ppm$^{-1}$: Inclusion of the scale factor may be appropriate. This discussion has been completely re-done.
   3. Ambient ug m$^{-3}$ vs STP ug sm$^{-3}$: Confirm that all values are reported at standard volume to compare like-to-like. We note this for each of the elevated site or aircraft studies, where it is reported.
   4. $PM_{2.5}$ vs $PM_1$: Studies using Aerosol Mass Spectrometers (e.g. Kleinmann et al., 2020, and Garofalo et al., 2019) will report non-refractory $PM_1$. So noted.

I acknowledge that choosing a convention will not have any bearing on the analysis, since this manuscript recommends performing the complete analysis for a particular location. Therefore, any definitions or units of PM will be consistent. However, uniformity in discussion of previous results and between the abstract and main text is appropriate. We have tried to maintain this uniformity as much as possible. We now consistently use the term $PM_{2.5}/CO$, rather than a more general PM/CO.

2. Ln 115: The authors state "Using the $PM_{2.5}/CO$ ratio to segregate the data, we find an improved correlation of PM and CO in the lower range of ratios, compared with using the HMS alone as an indicator (Figure 2)."

In Fig. 2, the $R^2$ values for the smoke days indicated by HMS smoke and PM2.5/CO>30 for the entire range seem comparable, while the $R^2$ values for the non-smoke days are less comparable, indicating the main difference between these methods is in the lower PM2.5 concentration range, well below the NAAQS. At these lower concentrations, the HMS smoke product is less likely to capture conditions at the surface and produces false negatives and positives for smoke-influence. To me, a major strength of the ratio method is the improved sensitivity and specificity in identifying smoke days at these lower concentrations. To highlight this, an SI figure explicitly showing the PM to CO correlations or an inset of $PM_{2.5}/CO$ vs CO that better shows this lower range would be helpful. Additionally, or alternatively, making the dots smaller or with some

transparency might allow the reader to better see the differences in the two methods at low concentrations in Fig. 2. Table 2 indicates that only a net change of 2 days between methods, but it seems that more than 2 dots have changed color between Fig. 2a and 2b. Can you add how many days switch categorization (and in which direction)?

I have added this line:

There are 612 days in the analysis.  200 have a positive HMS smoke identification and 220 have PM$_{2.5}$/CO ratios>30.   In total, 73 days with PM$_{2.5}$/CO ratios>30 do not have a positive HMS smoke identification.   As noted in Table 1, using a criteria of PM$_{2.5}$/CO>35, there are 27 days with identified smoke, but no HMS indication.

 I also suggest adding the NAAQS to Fig. 2 to show that both methods successfully identify exceedance days. Further explanation and slight tweaks to the figures for the low concentration data will further support the authors' assertation that the ΔPM$_{2.5}$/ΔCO method is generally a more robust indicator of surface smoke than satellite-based measurements.   Here are the two figures focusing on the lower end of the scale.   As noted, the correlation is much better using the PM$_{2.5}$/CO ratio, rather than the HMS data alone.  However, I don't really find that the figures are very compelling or showing anything that is not in the original figures.  So I choose not to include these.

[Figure]

[Figure]

I did redo Figure 2 to change the dot size and this seems to address the reviewers comment. New versions are below. These will be swapped out in the final manuscript.

[Figure]

A new development is the use of a Monte Carlo simulation to estimate $PM_{2.5}/CO$ ratios for smoke and urban influence separately in order to estimate the relative contribution of smoke to observed $PM_{2.5}$.

3. How sensitive are the Monte Carlo results to the chosen $PM_{2.5}$ and CO backgrounds and how do they compare to the non-smoke days (from either and both methods) from the Sparks site in 2019?

There is really a limited range of background values to use. CO might have a background from 0.1-0.2 ppm. $PM_{2.5}$ could have a background of 1-3 µg m$^{-3}$. From the observations, we find an observed $PM_{2.5}/CO=21.7$ for all $PM_{2.5}$ <10 µg m$^{-3}$

The table below shows the results of the Monte Carlo simulations results for the $PM_{2.5}$ range of 0-10 µg m$^{-3}$ using this range of background concentrations. There is not a lot of discrimination in the results based on the background concentrations. Given especially the significant significant uncertainty in the measurements at these lower concentrations I do not see a lot of value in adding this discussion.

| Background CO (ppm) / Background PM$_{2.5}$ (µg m$^{-3}$) | 0.1 | 0.2 |
|---|---|---|
| 1 | 22.4 | 14.2 |
| 2 | 28.1 | 17.7 |
| 3 | 33.7 | 21.0 |

4. The ozone discussion is limited and the numbers in line 190 do not seem to match the numbers in Table 1. Thank you. The values in Table 1 and at line 190 have been corrected.

Careful reading for grammatical errors and missing references (e.g. Briggs et al., 2016) will improve readability. Added missing references.

---

## Author Response (AR2)

**Responses in red.**

**6/22/22: Comments to the author**:
Thank you for your consideration of the referees' comments. I believe the comments have generally been adequately addressed and I am prepared to accept the manuscript subject to consideration of the following minor/technical points.

In response to referee 2's comment about sensitivity to background values, please add a sentence to the manuscript stating that a range of background CO and PM2.5 values were investigated and found to have minor impacts on the results.

**Added a sentence at line 171.**

Line 16: Please add the word "threshold" after "ratio."

**I substituted "threshold" for "ratio." here as I think it reads better and is clearer.**

Table 3: Please fix the formatting of the table to have units all on one line.

**Done.**

Please check the numbers in lines 134-136, 140-141, and Table 3 as these are inconsistent with each other. For instance, line 134 states 220 days with PM2.5/CO > 30 whereas table 3 states 198. Line 135 says 72 days with PM2.5/CO > 30 do not have a positive HMS smoke indication, whereas line 141 says 60 days.

**Thank you for identifying this issue. The problem came from inconsistencies in how days were counted (e.g. above or below 29.999, 30.000 or 30.999). To be consistent throughout the manuscript, I use a threshold of 30.0. This is now clearly stated and the values are updated in Table 3.**

Lines 231-232: Based on the analysis presented earlier, it seems that the satellite data perhaps has both false positives and negatives. From line 140-141, 53 days with HMS smoke but a PM2.5/CO ratio < 30 and 60 days with a ratio > 30 but no HMS smoke. Please clarify

**Yes. This is now discussed at line 137-140 and 238-239.**

Lines 233-234: I don't follow how the conclusion of no false positives for smoke identification was reached. I would think that would necessitate measurements of smoke specific tracers. Please elaborate on this conclusion.

**Agreed.  I have removed the statement about false positives here.**

Figure 2: Please fix the legend on the bottom panel such that the black points are PM2.5/CO < 30.

**Done.**

Figures 3 & 4: Please include units on the y-axis labels. I also recommend changing the ordering of the graphs such that the lines for the Rsmoke and Rurban values are drawn on top of the observation points. It is difficult to see how the Monte Carlo simulations represents the data at low PM2.5 concentrations. Please also consider using a color other than black for the observation data since the black Monte Carlo simulation results are difficult to see on top of the black dots.

**On both plots, I have changed the color of one line for better clarity.   I did my best to make the lines visible at the low end, but there is  very high density of points there.**

Figure 5: Please change the x-axis label to PM2.5/CO and include units.

**Done.**